# A Simple and Scalable Assay for Multiplexed Flow Cytometric Profiling of Surface Markers on Small Extracellular Vesicles

**DOI:** 10.3390/cells14130989

**Published:** 2025-06-28

**Authors:** Deborah Polignano, Valeria Barreca, Massimo Sanchez, Massimo Sargiacomo, Maria Luisa Fiani

**Affiliations:** 1National Centre of Global Health, Istituto Superiore di Sanità, 00161 Rome, Italy; deborah.polignano@iss.it (D.P.); barrecavaleria@gmail.com (V.B.); sargiacomomassimo@gmail.com (M.S.); 2Core Facilities, Istituto Superiore di Sanità, 00161 Rome, Italy; massimosanchez@libero.it

**Keywords:** small extracellular vesicles, sEV, exosomes, Bodipy FL C16, flow cytometry, tetraspanins, colocalization analysis

## Abstract

Extracellular vesicles (EVs), including small EVs (sEVs) such as exosomes, play crucial roles in intercellular communication and disease pathology. Their heterogeneous nature, shaped by cellular origin and activation state, requires precise and multiplexed profiling of surface markers for effective characterization. Despite recent advances, current analytical methods remain complex, costly, or inaccessible for routine laboratory use. Here, we present a simple and cost-effective flow cytometry-based assay for the multiplexed analysis of tetraspanin markers (CD63, CD81, CD9) on fluorescently labeled sEVs. Our method combines metabolic labeling with paraformaldehyde fixation and low-speed centrifugation using a benchtop centrifuge, enabling efficient removal of unbound antibodies and minimizing nonspecific signals while preserving vesicle integrity. Using either metabolically labeled exosomes or bulk sEVs stained with carboxyfluorescein succinimidyl ester (CFSE), we demonstrate robust recovery and accurate, semi-quantitative profiling of tetraspanin expression. The assay reveals substantial variability in tetraspanin distribution across different cell lines and does not require ultracentrifugation or immunocapture. Notably, this versatile and reproducible method supports high sEV recovery and is adaptable to additional protein markers. Its compatibility with standard laboratory equipment makes it a practical and scalable alternative to more complex techniques, expanding access to multiplex sEV analysis for both research and clinical applications.

## 1. Introduction

### 1.1. Background on EVs

Extracellular vesicles (EVs) are a heterogeneous population of lipid bilayer vesicles released into body fluids by nearly all cell types, playing a crucial role in intercellular communication under normal and pathological conditions. The classification of EVs is based on criteria such as size, density, lipid and protein composition, and subcellular origin [1,2]. Large extracellular vesicles (lEVs) have diameters larger than 200 nm, while small extracellular vesicles (sEVs) have diameters between 50 and 150 nm. Among sEVs, exosomes are formed as intraluminal vesicles within multivesicular bodies (MVBs), which subsequently fuse with the plasma membrane, whereas microvesicles or ectosomes bud directly from the plasma membrane. Recent data reveal significant heterogeneity among sEVs, exhibiting different markers based on the release mechanism and cellular origin. Due to their ability to transmit biological signals across tissues, sEVs are explored for therapeutic and diagnostic purposes.

### 1.2. Challenges in Current Methods for EVs Protein Profiling

Numerous studies have highlighted their potential as biomarkers for various diseases, including cancer, neurodegenerative disorders, and metabolic and cardiovascular diseases. However, profiling protein markers on EVs remains a challenging task and involves diverse methods such as proteomic analysis, flow cytometry, and immunocapture techniques [3]. Identified protein markers, including tetraspanins and other disease-related markers, hold potential for biomarker discovery and understanding disease biology. Furthermore, EVs analysis could be extremely valuable in studying their biogenesis and in understanding EVs functions, tumor heterogeneity and phenotypic changes occurring during therapy, and variations that occur concomitantly with tumoral changes. The heterogeneous composition of EVs thus requires the development of multiplexed EVs technologies. Several studies have proposed innovative methods for multiplexed EVs protein detection, including flow cytometry-based assays, immunocapture techniques, and sequencing-based protein analysis. However, current protocols often require expensive equipment and specialist knowledge. Furthermore, various isolation techniques, including ultracentrifugation, size exclusion chromatography, polymer-based precipitation, and immunocapture, can influence the protein composition of isolated EVs.

One method to analyze heterogeneous EVs involves immunostaining with fluorescent antibodies, as described in various studies [4,5,6,7]. This is often achieved through EVs capture or immobilization strategies, such as binding to a surface or to immunobeads conjugated to antibodies, typically directed at tetraspanins. While these strategies also facilitate the removal of unbound antibodies without requiring additional purification steps, a major drawback is the large sample volume needed to obtain sufficient EVs for analysis. Furthermore, methods relying on the capture of EVs using antibody-coated surfaces, like immunobeads, followed by the detection of specific proteins using labeled antibodies, enable the characterization of only sEVs carrying the selected protein marker, usually tetraspanins like CD63 [8,9,10,11]. To obtain more informative data, it becomes necessary to perform cross immunoprecipitation with beads carrying different antibodies or MACsPlex immunocapture assays, ensuring a more comprehensive characterization. Recently, a multiplex bead-based flow cytometric assay that allows for the detection of more than 30 EVs surface epitopes has been developed [12,13]. Although very promising, this method is very expensive. Therefore, there is a clear need for a sensitive and multiplex approach for sEVs marker detection that is both accessible and scalable.

### 1.3. The Rationale and Aim of Our Study

In our work, we leverage paraformaldehyde (PFA) fixation, a standard method in flow cytometry, to preserve vesicle integrity and surface antigenicity during staining. We hypothesized that PFA-induced protein crosslinking would increase vesicle rigidity, potentially altering sedimentation behavior and enabling the pelleting of fixed vesicles at lower g-forces. This, in turn, facilitates the removal of unbound antibodies without requiring additional purification steps such as filtration, ultracentrifugation, or size-exclusion chromatography.

Here we describe a simple and rapid method for analyzing the surface profiles of tetraspanins (CD9, CD63, and CD81) on PFA fixed fluorescent vesicles, which can be readily extended to other protein markers. Using metabolically labelled cell-derived sEVs that we extensively characterized as bona fide exosomes [14], we show substantial differences in the tetraspanin profiles of sEVs derived from various cell lines. The multiplex flow cytometric assay described here is broadly applicable in standard laboratory settings and enables reproducible, semi-quantitative detection of sEVs surface marker expression across various EV-containing sample types.

## 2. Materials and Methods

### 2.1. Cell Cultures

The human melanoma cells Mel501 were obtained by Istituto Nazionale Tumori (Milan, Italy) and authenticated according to a standard short tandem repeat-based genotyping at Ospedale Policlinico San Martino (Genova, Italy). The malignant human melanoma cell line A375M was kindly provided by R. Giavazzi (Istituto Mario Negri, Bergamo, Italy) and authenticated by standard short tandem repeat-based genotyping (Department of Oncology and Molecular Oncology, ISS, Rome, Italy). HeLa cervical adenocarcinoma cells (ATCC #CCL-2) and the human embryonic kidney HEK293T (ATCC #CRL-3216) were purchased from ATCC. Mel 501 cells were cultured in RPMI 1640; A375/M, HEK293T and HeLa cells were cultured in Dulbecco’s modified Eagle’s medium. All media were supplemented with 10% heat-inactivated fetal bovine serum (FBS), 100 units/mL penicillin, 100 µg/mL streptomycin, and 2 mM L-glutamine (complete media). Additional 1% MEM Non-Essential Aminoacids were added to A375M cell media. When required for CFSE staining of sEVs, FBS was replaced with Exosome Depleted FBS (EVs-depleted medium). All media, FBS and supplements were from EuroClone, S.p.A. Milan, Italy. Cell lines were grown at 37 °C, under 5% CO_2_, in humidified incubators and routinely tested for mycoplasma contamination using the Mycoplasma PCR Detection Kit (Applied Biological Materials (abm), Vancouver, BC, Canada, #G238).

### 2.2. Metabolic Labelling of sEVs

BODIPY FL C16 (Bodipy C16) (4,4-Difluoro-5,7-Dimethyl-4-Bora-3a,4a-Diaza-s-Indacene-3-Hexadecanoic Acid (ThermoFisher, Waltham, MA, USA, #D3821) was complexed with fatty acid-free bovine serum albumin (BSA), (Sigma-Aldrich, Saint Louis, MO, USA, #A8806) as described previously [15]. To isolate sEVs, 60–70% confluent monolayers of cells in exponential growth were incubated with 7 µM Bodipy C16 at 37 °C for 5 h (for Mel501) or 4 h (for HeLa, A375M, HEK293T) in medium containing antibiotics and glutamine and supplemented with 0.3% FBS (cell labelling medium). Cells were washed twice with Hank’s balanced salt solution (HBSS) containing 0.1% (*w*/*v*) fatty acid-free bovine serum albumin (H-BSA) to remove excess probe and further incubated for 24 h in complete culture medium. The conditioned medium was either immediately processed for sEVs isolation or stored at 4 °C for up to one week.

### 2.3. CFSE Labelling of sEVs

To prepare sEVs for generic fluorescent labeling, cells at 60–70% confluency were cultured in EV-depleted medium for 24 h. Conditioned medium was then collected and subjected to differential centrifugation as previously described [14] for sEVs isolation.

Freshly pelleted sEVs (100,000× *g*, hereafter referred to as 100 K sEVs) were labelled with 10 µM Carboxyfluorescein Diacetate Succinimidyl Ester (CFDA-SE, referred to as CFSE; ThermoFisher, Waltham, MA, USA, #C1157) for 30 min at RT in the dark. The labeling reaction was stopped by the addition of 100 mM L-glutamine. Fluorescent sEVs were then quantified by FC. CFDA-SE is a non-fluorescent molecule converted to fluorescent eCFSE (carboxyfluorescein succiminidyl ester) by intravesicular esterases. This helps to discriminate EVs from lipoproteins, as the latter do not contain esterase activity.

### 2.4. sEVs Isolation by Differential Ultracentrifugation

sEVs were isolated from 12 mL of conditioned medium collected from cells cultured in T75 flasks 24 h after fresh media addition. The medium was first centrifuged at 2000× *g* for 20 min at 4 °C to remove cells and large debris. The supernatant was then subjected to a second centrifugation at 10,000× *g* for 20 min at 4 °C to eliminate microvesicles and remaining particulates. The resulting supernatant was ultracentrifuged at 100,000× *g* for 90 min at 4 °C using an SW41-Ti swinging bucket rotor (Beckman Coulter, Milan, Italy). The pellet containing sEVs (100 K pellet) was washed in 12 mL of sterile PBS and centrifuged again at 100,000× *g* for 90 min under the same conditions. Final sEVs pellets were resuspended in 100–150 µL of PBS.

### 2.5. NanoFACS Analysis of Fluorescent sEVs

The quantification of fluorescent sEVs was evaluated as described with a conventional non-customized CytoFLEX LX flow cytometer (Beckman Coulter, Milan, Italy) by exploiting their fluorescent emission. To establish the region containing the sEVs, threshold and gain values were initially defined on the appropriate fluorescent channel with a bandpass of 525/40 by using fluorescent beads of known sizes ranging from 100 to 900 nm (Megamix Plus FSC/SSC, BioCytex, Marseille, France, #7802/7803) [13]. The instrument was then set to include fluorescent-size reference beads smaller than 100 nm. To obtain the number of sEVs, the Cytoflex LX instrument utilized a sample peristaltic pump capable of calibrating the volume delivery of the sample, allowing for absolute EVs counts without the need for bead-based calibration.

### 2.6. Bodipy-Exo and CFSE-sEVs Immunostaining

To determine colocalization between Bodipy-exo and tetraspanins, freshly pelleted Bodipy-exo isolated by ultracentrifugation were either first incubated with anti-tetraspanins antibodies for 45 min at room temperature (RT) in a HulaMixer Sample Mixer in the dark or were pre-fixed with 4% Paraformaldehyde in PBS for 30 min at RT before the antibody staining step. The assay volume was 25 or 100 µL, as indicated in the figure legends. Non-fixed stained samples were subsequently fixed with 4% PFA in PBS for 30 min at RT and washed with PBS at 500× *g* for 20 min at 4 °C on a benchtop centrifuge (Ole Dich Cooling Centrifuge Type 157.MP). The supernatant was discarded, and pelleted Bodipy-exo was resuspended in 300 µL PBS. Alternatively, after incubation with the antibodies, 200 µL of PBS + 2% FBS was added to Bodipy-exo and samples were washed in Nanosep filters with a 300 kDa cut-off (Avantor, Radnor, PA, USA #5168531) and then centrifuged at 10,000× *g* for 20 min at 4 °C. Bodipy-exo was then resuspended in 300 µL of PBS. The colocalization percentage of Bodipy-exo with BV421 anti-CD63, APC anti-CD81, and PE anti-CD9 was determined by FC. The following antibodies were used: BV421 anti-CD63 antibody (Becton Dickinson, EastRutherford, NJ, USA, #740080), APC anti-CD81 antibody (Becton Dickinson, EastRutherford, NJ, USA, #551112), PE anti-CD9 antibody (Becton Dickinson, EastRutherford, NJ, USA, #555372). Additionally, as isotype controls, we used antibody IgG1,κ conjugated to either BV421 (Becton Dickinson, EastRutherford, NJ, USA, #569394), APC (Becton Dickinson, EastRutherford, NJ, USA, #555751), or PE (Becton Dickinson, EastRutherford, NJ, USA, #555749) in filtered PBS with 2% FBS. Immunostaining was carried out as above. Data were acquired using a CytoFLEX LX flow cytometer (Beckman Coulter Life Sciences), equipped with five lasers (355, 405, 488, 561, and 640 nm wavelengths). As we previously described [15,16], we set the acquisition threshold on the B525-FITC fluorescence channel, corresponding to the emission of Bodipy-labelled lipids. This thresholding approach is specifically designed to eliminate background noise and exclude events not emitting in the B525-FITC range, thereby enriching for events that are either single-positive for Bodipy (Bodipy-exo) or double-positive for Bodipy and anti-tetraspanin antibodies. Events falling below this fluorescence threshold, including double-negative vesicles or vesicles positive only for the antibody label, were not acquired and therefore do not appear in the flow cytometry plots. All samples have been recorded at a medium flow rate of 30 µL/minute for a total of 30 µL. Fluorescent signals were collected as follows: FITC was measured in B525-FITC channel (525/40 nm filter), Brilliant Violet 421 (BV421) was measured in 450-PB channel (450/45 nm filter), Phycoerythrin (PE) was detected in Y585-PE channel (585/42 nm filter), and Allophycocyanin (APC) was detected in R660-APC channel (660/10 nm filter). Data analysis was performed using CytExpert v2.4 and FlowJo v10.10 software. Detergent lysis controls were prepared for Bodipy-exo and CSFE-sEVs labelled with anti CD63, CD81, and CD9 antibodies by incubating the samples with 60 mM n-octyl-β-D-glucoside (Sigma-Aldrich, Saint Louis, MO, USA, #850511P) (OG), a nonionic detergent widely used for membrane protein solubilization, in order to discriminate between protein aggregates and membrane.

### 2.7. Western Blot

EVs proteins were separated by 12% sodium dodecyl sulphate (SDS) polyacrilamide gel electrophoresis (PAGE) under reducing conditions, with the exception of samples used to detect CD63, CD81, and CD9, and transferred on 0.22 µm nitrocellulose membranes (Bio-Rad, Hercules, CA, USA). Membranes were blocked for 5 min at RT in EveryBlot Blocking Buffer (Bio-Rad, Hercules, CA, USA) and then incubated for 1 h at RT with different primary antibodies: mouse anti-CD63 (Becton Dickinson, EastRutherford, NJ, USA, #556019, 1:300), mouse anti-CD81 (Becton Dickinson, EastRutherford, NJ, USA, #555675, 1:300), and rabbit anti-CD9 (Cell Signaling Technology (CST), Danvers, Massachusetts, USA, #13403, 1:300). Horseradish peroxidase (HRP)-conjugated secondary antibodies were used 1:3000 for 1 h at RT in EveryBlot. Western blots were developed using Clarity Western ECL Substrate (Bio-Rad, Hercules, CA, USA). The presented immunoblots are representative of at least 3 independent experiments. Band analysis was performed using ImageLab software 6.1 from Bio-Rad, Hercules, CA, USA.

### 2.8. Statistical Analysis

Statistical analyses in this study were conducted with Graphpad Prism 9.5 software. The data are presented as the mean ± SD from at least three independent experiments. Individual group statistical comparisons were analyzed by the two-tailed Student t test. Differences between groups were analyzed by one-way ANOVA with Tukey’s multiple comparison post hoc test. *p*-values are represented as * *p* ≤ 0.05, ** *p* ≤ 0.01, *** *p* ≤ 0.001, and **** *p* ≤ 0.0001. A *p*-value ≤ 0.05 was considered statistically significant.

## 3. Results

### 3.1. Comparison of Different Strategies for Bodipy-Exo Immunostaining

We previously showed that by using BODIPY FL C16 (Bodipy C16), a fluorescent palmitic acid analogue that is rapidly transformed by cells into the phospholipid components of sEVs membranes, we can obtain a fluorescent subpopulation of small exosomes (Bodipy-exo) characterized as recommended by MISEV2023 guidelines [3], whose number and fluorescence intensity can be precisely analyzed by flow cytometry (FC) [12]. By taking advantage of this methodology, we set up a simple protocol illustrated in Figure 1a–c to determine the tetraspanins profile of paraformaldehyde-fixed Bodipy-exo.

In establishing our protocol, we conducted a comparison of Bodipy-exo colocalization with CD63, CD81, and CD9 tetraspanins by implementing the paraformaldehyde fixation step either before adding the fluorescent primary antibodies and isotype controls (Figure 1a) or after 45 min incubation with antibodies (Figure 1b). This comparison was crucial to rule out any modification of epitopes that the antibodies recognize on Bodipy-exo due to the fixation step. Direct immunostaining of sEVs in solution implies a subsequent step for removing antibodies once the staining process is complete. The presence of unbound antibodies or aggregates can result in false-positive signals when analyzed by FC [4,6,17,18]. To minimize this, we washed PFA-fixed Bodipy-exo with PBS using low-speed centrifugation (500× *g*) on a standard benchtop centrifuge. Consequently, we washed the fixed Bodipy-exo with PBS through centrifugation at low speed (500× *g*) using a benchtop centrifuge. By quantifying Bodipy-exo before and after centrifugation, we confirmed that PFA-fixed vesicles were efficiently recovered under these conditions, with yields comparable to those obtained using ultracentrifugation or filtration. This supports the hypothesis that fixation-induced rigidity and/or aggregation enhances vesicle sedimentation at lower centrifugal forces. For comparison, we omitted the fixation step and, after the antibody incubation, concentrated and washed the vesicles using 300 kDa filters, as previously described [19] (Figure 1c). FC analysis of colocalization in the different conditions revealed that performing the fixation step after the incubation of Bodipy-exo with the antibodies showed no significant difference compared to untreated exosomes washed and concentrated using filters. However, in the case of first fixing the vesicles with PFA, some epitope modification may occur. This resulted in an underestimation of the percentage of colocalization for CD63 and an overestimation for CD9 (Figure 1d). The protocol involving immunostaining first, followed by the fixing step (Figure 1b), was subsequently chosen for further characterization.

### 3.2. Optimization of Bodipy-Exo and Tetraspanin Antibody Concentration

The optimal concentration of individual antibodies is crucial for achieving high-quality staining while minimizing interference from the isotype control signal [20,21]. This is particularly important in multiplexed FC analysis, where different antibodies conjugated to distinct fluorophores are used simultaneously. Before proceeding with the validation of our assay, we conducted a control experiment testing Bodipy-exo coincidence occurrence through serial dilution of Bodipy-exo. Sample acquisition was performed taking into consideration both sample concentration and the number of events per second acquired at a medium flow rate of 30 µL/min. At high sample concentrations, the linear relationship between serial dilution and events per second was reduced (R^2^ = 0.93) (Appendix A). This reduction likely results from an increased event abort rate and the clustering of multiple events, leading to their detection as single events (swarm effect). To mitigate these issues, we selected conditions where the event abort rate remained below 5%, yielding a more robust regression curve (R^2^ = 0.96). These parameters were subsequently applied in all further analyses to ensure accurate and reliable quantification of sEVs [16].

As the next step, we incubated varying amounts of Bodipy-exo, ranging from 2.5 × 10^6^ to 1 × 10^7^, with antibody concentrations between 2.5 and 10 µg/mL (Figure 2a). To minimize antibody consumption, we reduced the assay volume for the antibody incubation step to a fixed volume of 25 µL. The percentage of tetraspanin-positive Bodipy-exo was determined by FC at each concentration of antibody and Bodipy-exo amount. As shown in Figure 2a, the maximum percentage of CD63^+^ Bodipy-exo was observed at an antibody concentration of 10 μg/mL and 2.5 × 10^6^ Bodipy-exo, whereas CD81 and CD9 antibodies reached saturation at 5 μg/mL and 2.5 μg/mL, respectively, regardless of the amount of Bodipy-exo (Figure 2a). Isotype controls were performed for each antibody concentration to determine the threshold in order to reduce the background noise (Appendix A).

As observed in Appendix A, the background noise is almost negligible when the same concentration of antibody and isotype is used for CD63 and CD81. However, the PE-conjugated isotype shows an increase in background noise (highlighted by the red rectangle in Appendix A), which correlates with the amount of isotype.

The variability in isotype controls is a well-known issue influenced by different factors such as antibody supplier, fluorochrome-to-protein ratio, antibody concentration, propensity for aggregation, and antibody subclass [22,23,24]. This issue is particularly relevant for sEVs, where the low number of antigens can make it challenging to distinguish between true positive signals and background noise. When we compared the data obtained in a 25 µL assay volume with previous data in a 100 µL assay volume (Figure 1d), we observed that performing the antibody staining step in the smaller volume led to a marked reduction in Bodipy-exo staining with the anti-CD63 antibody. Specifically, with 10 µg/mL of antibody and 1 × 10^7^ Bodipy-exo in 100 µL, the percentage of CD63^+^ Bodipy-exo was approximately four times higher (Figure 1d) compared to the 25 µL assay volume. Increasing the anti CD63 concentration to 20 µg/mL in a 25 µL assay volume did not significantly increase the percentage of CD63^+^ Bodipy-exo (Figure 2a). In contrast, the anti-CD81 and CD9 antibody staining did not seem to be affected.

To assess the effect of the assay volume on anti CD63 staining, we repeated the optimization experiment by testing different amounts of Bodipy-exo and concentrations of anti CD63 antibodies in a 100 µL assay volume. As shown in Figure 2b, the percentage of CD63^+^ Bodipy-exo at an antibody concentration of 10 μg/mL matched the results we previously obtained (Figure 1d) (Figure 2b and Appendix A). One possible explanation is that the extensive glycosylation of CD63 affects antibody recognition when the assay volume is too small and/or the number of sEVs is too high. Despite the preliminary data for CD81 and CD9 (Figure 1d) not seeming to indicate a difference in the percentage of Bodipy-exo colocalization due to assay volume, we repeated the experiment by using a fixed Bodipy-exo concentration (1 × 10^7^ Bodipy-exo) and the minimum antibody concentration giving saturation (10 μg/mL for CD63, 5 µg/mL for CD81, and 2.5 µg/mL for CD9) (Figure 2a,b). As shown in Figure 2c, there is no significant difference in the percentage of CD81- and CD9-positive Bodipy-exo when the incubation is carried out in a volume of 25 or 100 µL as compared with CD63. Therefore, in all subsequent experiments, an assay volume of 100 µL was chosen for all CD63 experiments, and an assay volume of 25 µL was chosen for CD81 and CD9, which also allowed us to perform a multiplex analysis for these tetraspanins.

The mean fluorescent intensity (MFI) (Figure 2d), which reflects the expression of each tetraspanin, showed no significant changes for any combination of CD63 and CD81 antibody concentrations and Bodipy-exo numbers. However, the MFI for the CD9 antibody exhibited increased variability, as also indicated in the flow cytometry dot plots (Appendix A). These plots show differing profiles for the MFI of the PE-labeled anti-CD9 antibody and its relative isotype compared to the MFI of BV421 and APC-labeled antibodies. The MFI variability shown by PE-conjugated anti-CD9 antibody could be to the increased unspecific background noise observed at higher concentrations of the PE-conjugated isotype (Appendix A, red rectangle). Based on these results, which helped identify the optimal antibody concentrations to minimize nonspecific signals, we selected 1 × 10^7^ input Bodipy-exo and 10 μg/mL antibody in 100 μL assay volume for CD63 and 1 × 10^7^ input Bodipy-exo along with antibody concentrations of 5 μg/mL for CD81 and 2.5 μg/mL for CD9 in 25 μL assay volume for further comparative analyses.

### 3.3. Validation of Signal Specificity and Background Reduction

Finally, to confirm the specificity of colocalization of tetraspanin antibodies to Bodipy-exo and to quantify the contribution of unbound reagents to background noise, we performed control experiments with unlabelled sEVs or PBS as recommended by the MIFlowCyt-EV framework [25].

Isotype controls for each tetraspanin antibody were included to determine the gating strategy and establish the positivity threshold for Bodipy-exo (Figure 3a). Events falling below this fluorescence threshold, including double-negative vesicles or vesicles positive only for the antibody signal, were not acquired and therefore do not appear in the plots. This accounts for the absence of signals in the upper and lower left quadrants. In contrast, the right quadrants reflect true fluorescent events. The minimal signals occasionally detected in control samples (e.g., PBS or unlabelled vesicles) are likely attributable to vesicle autofluorescence or rare nonspecific events. Using these acquisition settings, the incubation of Bodipy-exo with anti-tetraspanin antibodies yielded distinct double-positive (DP) sEVs populations, which were used for comparative analysis. To assess background noise and the contribution of antibody aggregates, we also incubated unlabeled sEVs or PBS with anti-tetraspanin antibodies. This led to a substantial reduction in the total number of detected positive events (Figure 3b). These results demonstrate that fluorescently stained vesicles can be clearly distinguished from background signals and supports the specificity of anti-tetraspanin antibody binding to Bodipy-exo.

### 3.4. CFSE Staining Does Not Interfere with Tetraspanins Antibody Labelling

To evaluate the general applicability of our method to different sources of sEVs, we stained sEVs with CFDA-SE, a membrane-permeable protein-binding dye converted to fluorescent CFSE by esterases present in the lumen of EVs. CFSE labelling is one of the most used labelling methods to obtain fluorescently labelled sEVs [7,26,27] and has also been employed to study EVs internalization and content transfer in vitro [28]. To this end, we labelled total sEVs pellets with CFSE. As shown in Figure 4a, colocalization analysis revealed significant differences in the percentage of CD63- and CD81-positive vesicles between Bodipy-exo and total CFSE-sEVs. This finding is consistent with our previous results, which demonstrated that Bodipy-exo is a bona fide exosome derived from intracellular compartments and is completely dissolved when treated with detergent [14]. In contrast, CFSE-sEVs, which represent the total sEVs population including microvesicles, ectosomes, and protein aggregates, exhibited a lower percentage of CD63- and CD81-positive vesicles. This result aligns with our previous observations that CFSE-labeled vesicles, when loaded onto an iodixanol density gradient, separate into two distinct peaks: a lower-density peak, which is completely soluble in detergent and falls within the same range as Bodipy-exo, and a higher-density peak that is only partially soluble in detergent, likely consisting of protein aggregates. No significant differences were observed for CD9. We next performed a detergent lysis step to PFA-fixed and antibody-stained Bodipy-exo or CFSE-sEVs. As shown in Figure 4b, almost all detected fluorescence-positive events in the FITC gate were absent after detergent lysis of Bodipy-exo, whereas CFSE-sEVs events were still detectable, confirming that detergent-resistant CFSE-sEVs likely consisted of protein aggregates and not membrane-enclosed vesicle [24,29].

Our results suggest that CFSE per se does not interfere with antibody binding, and the staining procedure described here can be successfully applied to bulk sEVs preparations, allowing fluorescent labelling after the isolation step. However, there remains the possibility that detergent-insoluble contaminants may bind to the antibodies, potentially increasing the risk of false-positive results.

### 3.5. Characterization of Tetraspanins Expressions in sEVs from Different Cell Lines

Several reports, primarily through mass spectrometry analyses of bulk EVs, have shown that sEVs of endosomal origin are enriched on their surface in CD63, CD81, and CD9 tetraspanins, although in different combinations and amounts and depending on the cell type [9,10,11,30,31,32]. Furthermore, the presence of specific tetraspanins in EVs has been associated with distinct cellular and body fluid signatures [13].

To explore differences among sEVs from various cell lines, we next applied our methodology to investigate the expression profiles of the tetraspanin markers in sEVs derived from four cell lines: Mel501, A375M, HEK293, and HeLa. Since we have already shown [14] that different cell types secrete different amounts of Bodipy-exo, in this study, we refer to the amount of Bodipy-exo as determined by FC irrespective of the cell source. As shown in Figure 5a, there is a heterogeneous distribution of individual biomarkers across sEVs populations from all cell lines. CD63^+^ Bodipy-exo was the most abundant compared to CD81- and CD9-positive Bodipy-exo, although there were significant differences among sEVs. Conversely, the percentage of CD81^+^ Bodipy-exo was more homogeneous among the different tested cell lines. CD9^+^ Bodipy-exo showed significant variability among the cell lines. Notably, HeLa cells had a considerably higher amount of CD9^+^ Bodipy-exo, whereas other cell lines showed lower expression of CD9^+^ Bodipy-exo, with significant differences observed among them. We could then determine the percent of positive Bodipy-exo for each tetraspanins in Mel 501, which includes CD63 (16.8%), CD81 (7.2%) and CD9 (3.0%); A375M, CD63 (10.8%), CD81 (6.6%) and CD9 (6.0%); HEK293T CD63 (8.8%), CD81 (8.5%) and CD9 (2.9%); HeLa CD63 (15.5%), CD81 (8.1%). and CD9 (14.6%).

The analysis of tetraspanin-positive Bodipy-exo is, however, independent of the protein copy number expressing Bodipy-exo. The fluorescence intensity of a fluorescent antibody-stained Bodipy-exo should provide information about the expression levels of the tetraspanins on the surface of Bodipy-exo. Therefore, we next determined the mean fluorescent intensity (MFI) of the antibody-stained Bodipy-exo for each tetraspanin marker. The fluorescence signal of each dye was detected by the corresponding detector [33]. The results presented in Figure 5b indicate that the number of tetraspanin copies on Bodipy-exo varies significantly among different cell lines, with particularly striking differences observed for CD9. Although only about 3% of Mel501 and HEK293T Bodipy-exo were CD9-positive, the MFI of CD9-positive Bodipy-exo was significantly higher in Mel501 compared to HEK293T (Figure 5b). Interestingly, HeLa Bodipy-exo, which had the highest percentage of CD9-positive Bodipy-exo, exhibited a lower MFI, suggesting that these vesicles contain fewer copies of the CD9 marker compared to those from Mel501 and A375M. Since Bodipy-exo represents only a subset of the total sEVs population, we next determined if the MFI for each tetraspanins on Bodipy-exo correlates with the amount of marker on sEVs determined by a bulk method such as Western blotting. The results in Figure 5c,d show that the CD63 content in total sEVs from different cell lines correlates with both the MFI and the percentage of CD63^+^ Bodipy-exo. In contrast, notable discrepancies were observed for CD81 and CD9. Specifically, although total sEVs from Mel501 and HeLa cells exhibited lower CD81 content (Figure 5c), the MFI indicated a higher number of CD81 copies on each Bodipy-exo. Similarly, a low total content was observed for CD9 in Mel501 cells, suggesting that this tetraspanin is present on a small subset of Bodipy-exo (Figure 5a) but with a higher density. Altogether, these results confirm that CD63, CD81, and CD9 tetraspanins are present on the surface of Bodipy-exo, although in different combinations and amounts.

## 4. Discussion

Small vesicles secreted from cells carry biomolecular cargos reflecting their biogenesis pathway and their cells of origin. Therefore, the surface marker profile of extracellular vesicles (EVs) is likely dependent on the cell source, the cell’s activation status, and multiple other parameters. This complexity underscores the importance of protein profiling for characterizing the heterogeneous populations of EVs subtypes [9,14,34], understanding their role in pathological conditions [19,35,36,37,38], and identifying novel markers [39,40,41,42]. Despite recent advancements in achieving quantitative and qualitative analysis of vesicle surface marker profiles [4,24,43,44,45], a reliable and affordable flow cytometry-based approach that does not require specialized equipment and expertise is still lacking. The small size of EVs makes single-EV analysis technically challenging, and several factors can affect the results, including the bulk nature of the analysis and the methods used for purification and staining. However, even with improved isolation techniques, the inherent small size of sEVs (typically ranging from 30 to 150 nm) poses a technical challenge for effective visualization and quantification during EVs immunostaining procedures, as most EVs fall below the detection sensitivity of commercially available flow cytometers. We recently showed that by metabolically labelling sEVs with a Bodipy-labelled palmitic acid, we obtain a discrete population of fluorescent exosomes homogeneous in size that can be precisely quantified by FC [14].

Here, we describe a simple and reliable method for multiplex analysis of tetraspanin markers on sEVs membrane surfaces, utilizing a PFA fixation step on antibody-stained fluorescent sEVs. This step allows for the rapid removal of unbound antibodies using a benchtop centrifuge, which is critical as aggregated antibodies can generate nonspecific signals [4,19]. To ensure that PFA fixation would not interfere with epitope binding, we performed control experiments, which showed that applying the fixation step after antibody staining resulted in the same percentage of tetraspanin-positive vesicles as observed with unfixed sEVs, where unbound antibodies were removed by filtration. An additional advantage of our procedure over methods such as filtration, ultracentrifugation, SEC, or density gradient separation is the higher recovery of stained sEVs, allowing for the use of a smaller starting quantity of sEVs (2.5 × 10^6^ Bodipy-exo or CFSE-stained sEVs). To enhance the specificity of our assay, detection of antibody-stained sEVs was limited to fluorescently labeled vesicles, such as Bodipy-exo and CFSE vesicles, by applying a gating strategy to minimize vesicle aggregates and reduce electronic background noise [15]. The choice of antibodies is also a critical factor, as they must target proteins present at low levels on the surface of sEVs and be sufficiently bright to enhance the detection threshold of the flow cytometer. This is essential for accurately phenotyping a small number of proteins on the surface of sEVs. To validate our methodology, we used the tetraspanins CD63, CD81, and CD9, which have been identified in some sEVs subpopulations in different combinations and are used as targets to immuno-isolate sEVs. To define the most optimal incubation protocols, control titrations were performed for each antibody to determine the optimal concentration of antibody and sEVs. Interestingly, while CD81 and CD9 staining conditions were similar, effective CD63 staining required an increased assay volume and a lower number of sEVs. This effect may be attributed to the extensive glycosylation of the CD63 molecule. Steric hindrance caused by bulky carbohydrate moieties can physically block antibody access to specific protein epitopes, highlighting the importance of conducting appropriate control experiments to optimize staining conditions and improve the accuracy of detection [46].

The surface marker profile of EVs is influenced by various factors including the cell source, the biogenesis pathway, and the activation status of the cell. This complexity underscores the importance of protein profiling to characterize the heterogeneous populations of EVs subtypes. Notably, the expression of tetraspanins, such as CD9, CD63, and CD8, commonly recognized as markers for sEVs, can vary significantly depending on the cellular origin of the vesicles. In the current study, we demonstrate the feasibility of our method for analyzing a homogeneous subpopulation of sEVs derived from different cell lines. In line with previous research, we observed a remarkable heterogeneity among various cell lines in CD63- and CD9-positive Bodipy-exo, whereas the percentage of CD81-positive Bodipy-exo appeared consistent across the tested cell lines. Although tetraspanins are well-known sEV-associated markers, recent research has shown that their expression varies across sEVs subclasses and is unevenly distributed among individual extracellular vesicles. Interestingly, our results reveal significant differences in the mean fluorescent intensity of tetraspanin-positive Bodipy-exo from various cell sources, suggesting that tetraspanin expression on vesicles does not necessarily correlate with the proportion of tetraspanin-positive vesicles but appears to be cell-specific. Notably, CD81 and CD9-positive vesicles derived from melanoma Mel501 cells, although not very abundant, exhibited a high level of protein expression (MFI) on single Bodipy-exo, indicating that the vesicles, although not very represented, are enriched in these tetraspanins. Small EVs secreted by cells differ in their subcellular origin, cargo, and surface markers, resulting in a highly heterogeneous population. As expected, when we assessed the tetraspanin content in total sEVs using a bulk method, such as Western blotting, the results did not correlate with the proportion of tetraspanin-positive Bodipy-exo or their expression levels on the fluorescent vesicles. Notably, only CD63, a marker associated with sEVs of intracellular origin and crucial for their biogenesis and function, showed consistent correlation between its expression level on each Bodipy-exo and the total sEVs content across all tested cell lines. In this study, we demonstrated that protein profiling can also be performed on bulk sEVs stained with CFSE.

To validate our method, we focused on determining the fraction of tetraspanin-positive fluorescent sEVs. However, as reported by other authors, the choice of EVs labeling dye must be carefully considered, as not all dyes exhibit the same effectiveness in labeling specific EVs subtypes. Therefore, appropriate control experiments should be included to assess dye suitability and ensure accurate interpretation of results. Our approach can be readily extended to analyze the proportions of all possible combinations of double-positive vesicles. The findings presented here provide valuable insights into EVs characterization. Notably, this versatile and reproducible method supports high sEVs recovery and can be readily adapted to detect additional protein markers beyond the tetraspanins tested in this study. By combining metabolic labeling of sEVs with a streamlined protein-profiling assay, we successfully identified cell-specific protein signatures on exosomes.

In conclusion, this method enables laboratories equipped with standard flow cytometry to detect surface markers on EVs, including low-abundance proteins. While some technical expertise remains necessary, as is typical for flow cytometry-based analyses, the overall workflow is simplified and accessible, avoiding the need for ultracentrifugation or specialized equipment. Overall, we present a simple, reliable, and highly flexible approach for protein profiling across different EVs sources, which may benefit both researchers and translational applications in clinical diagnostics.

## Figures and Tables

**Figure 1 cells-14-00989-f001:**
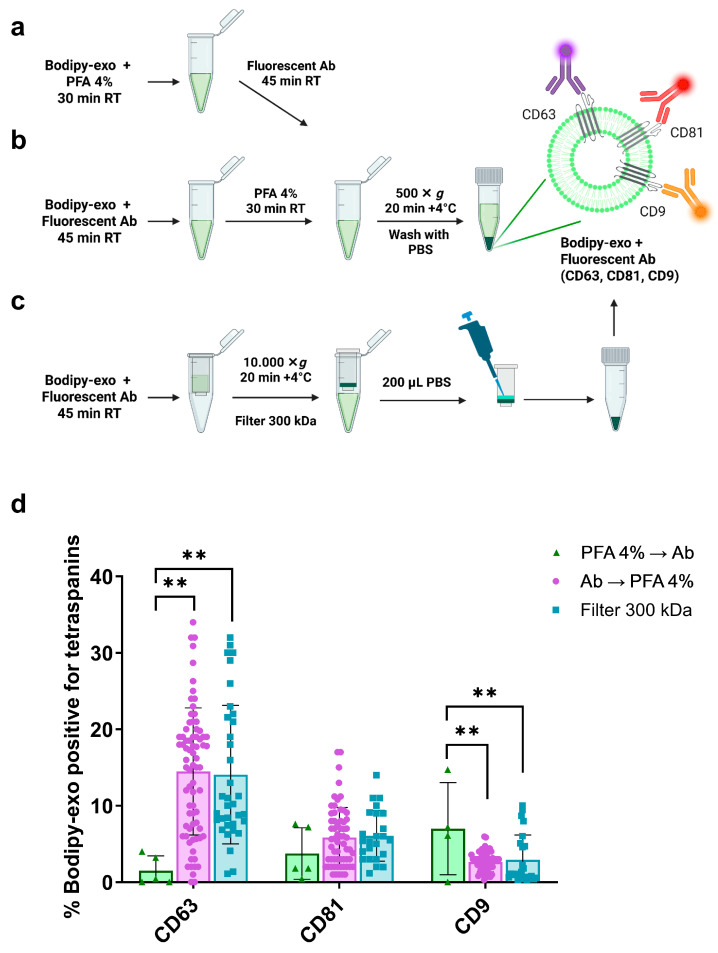
(**a**–**c**) Schematic representation of the experimental strategy. (**d**) Comparative analysis of PFA fixed and filter washed Bodipy-exo for tetraspanins colocalization using flow cytometry. The assay volume for tetraspanin antibody staining was 100 µL, with 107 Bodipy-exo from Mel501 cells and an antibody concentration of 10 µg/mL for CD63, 5 µg/mL for CD81, and 2.5 µg/mL for CD9. Data show mean ± SD of at least 4 independent experiments. Statistical significance was determined by one-way ANOVA and Tukey’s multiple comparison test. *p* values < 0.05 were considered statistically significant. ** *p* < 0.01.

**Figure 2 cells-14-00989-f002:**
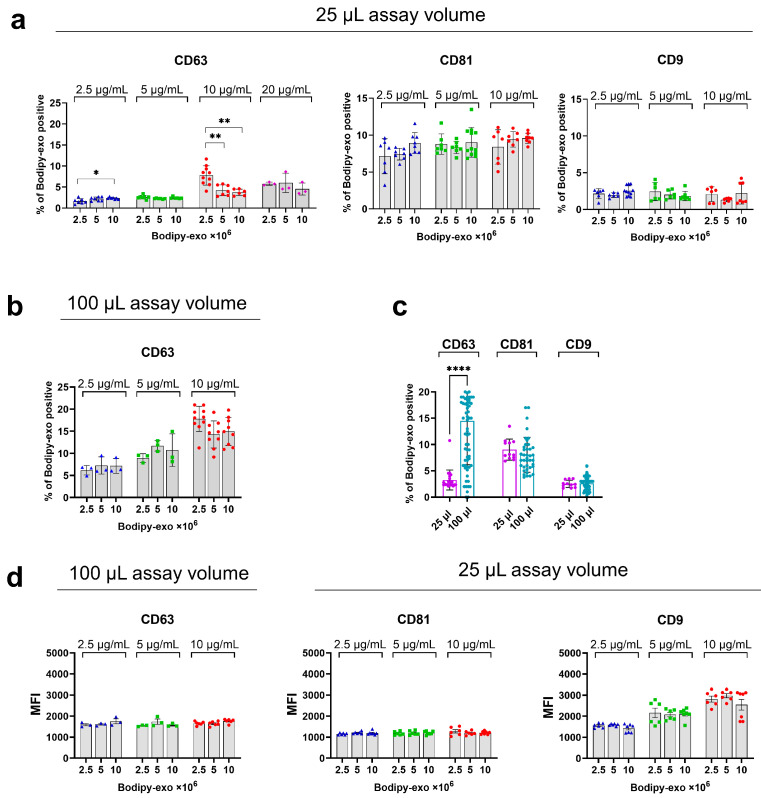
Titration of antibody concentration and Bodipy-exo from Mel501 cells amounts. (**a**) Antibody titration for CD63, CD81, and CD9 ranging from 2.5 to 20 µg/mL and different amounts of Bodipy-exo, ranging from 2.5 × 10^6^ to 10^7^, in an assay volume of 25 μL. Data show mean ± SD of at least 3 independent experiments. Statistical significance was determined by one-way ANOVA and Tukey’s multiple comparison test. (**b**) Antibody titration for CD63 tetraspanin in an assay volume of 100 μL and (**c**) comparison of 10 µg/mLCD63, 5 µg/mL CD81, and 2.5 µg/mL CD9 incubation with 1 × 10^7^ Bodipy-exo in 25 μL and 100 μL assay volume. Data show mean ± SD of at least 3 independent experiments. Statistical analysis was performed using paired Student’s *t*-test. (**d**) Mean fluorescent intensity (MFI) of the respective fluorescent signals performed in an assay volume of 100 μL for CD63 and 25 μL for CD81 and CD9. Data show mean ± SD of at least 4 independent experiments. *p* values < 0.05 were considered statistically significant; * *p* < 0.05, ** *p* < 0.01, and **** *p* < 0.0001.

**Figure 3 cells-14-00989-f003:**
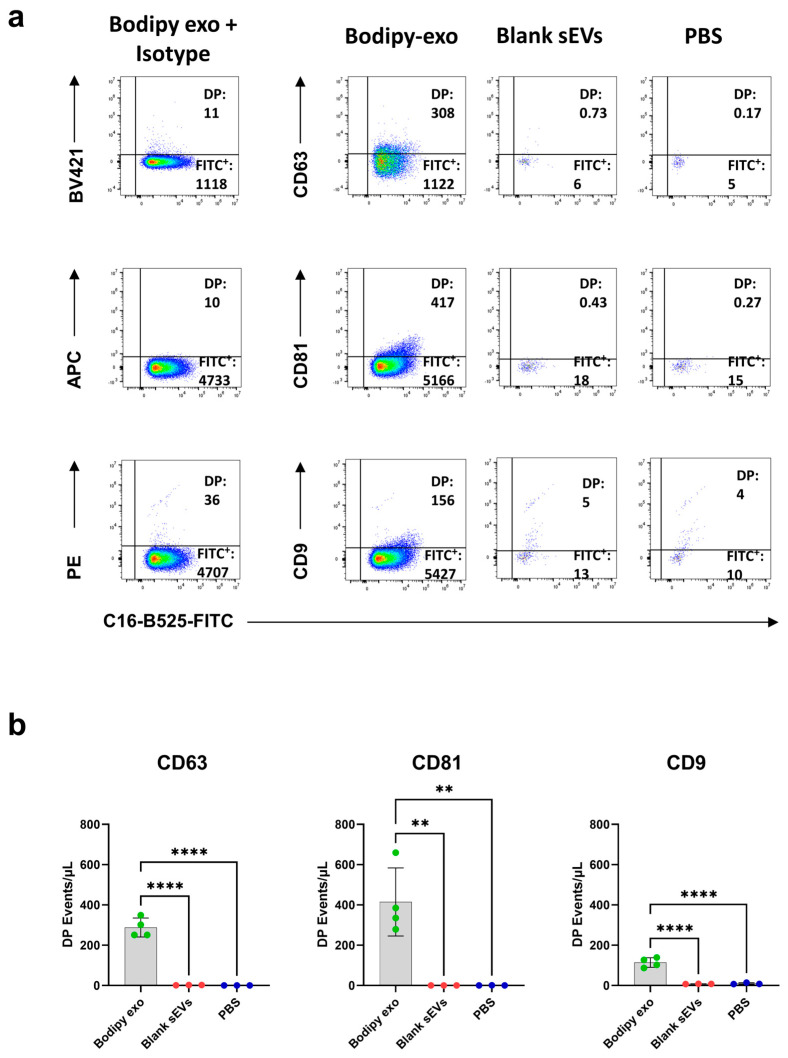
Staining controls: (**a**) 1 × 10^7^ Bodipy-exo, unlabelled sEVs derived from the same cell number yielding 1 × 10^7^ Bodipy-exo or PBS were incubated with fluorescent antibodies anti-CD63 (10 µg/mL), CD81 (5 µg/mL), and CD9 (2.5 µg/mL). The number of double positive (DP) populations expressed as events/μL of Bodipy-exo were compared with the DP events/μL on unlabeled sEVs and PBS incubated with BV421 anti-CD63, APC anti-CD81, and PE anti-CD9 conjugated antibodies. The dot plots in the first column on the left show the Bodipy-exo incubated with the related isotype controls that was used to set the gate of the DP populations. The double positive (DP) events for Bodipy and each tetraspanin have been identified by setting the threshold on the FITC channel (band pass 525/40) and acquiring 30 μL with a medium flow rate (30 μL/min) for all samples. Plots are representative of n ≥ 3 independent experiments. (**b**) Graphical representations of quantification of DP events/μL of Bodipy-exo, unlabelled sEVs, or PBS. Data are expressed as mean ± SD (n ≥ 3). Statistical analysis was performed using one-way ANOVA and Tukey’s multiple comparison test. *p* values < 0.05 were considered statistically significant; ** *p* < 0.01 and **** *p* < 0.0001.

**Figure 4 cells-14-00989-f004:**
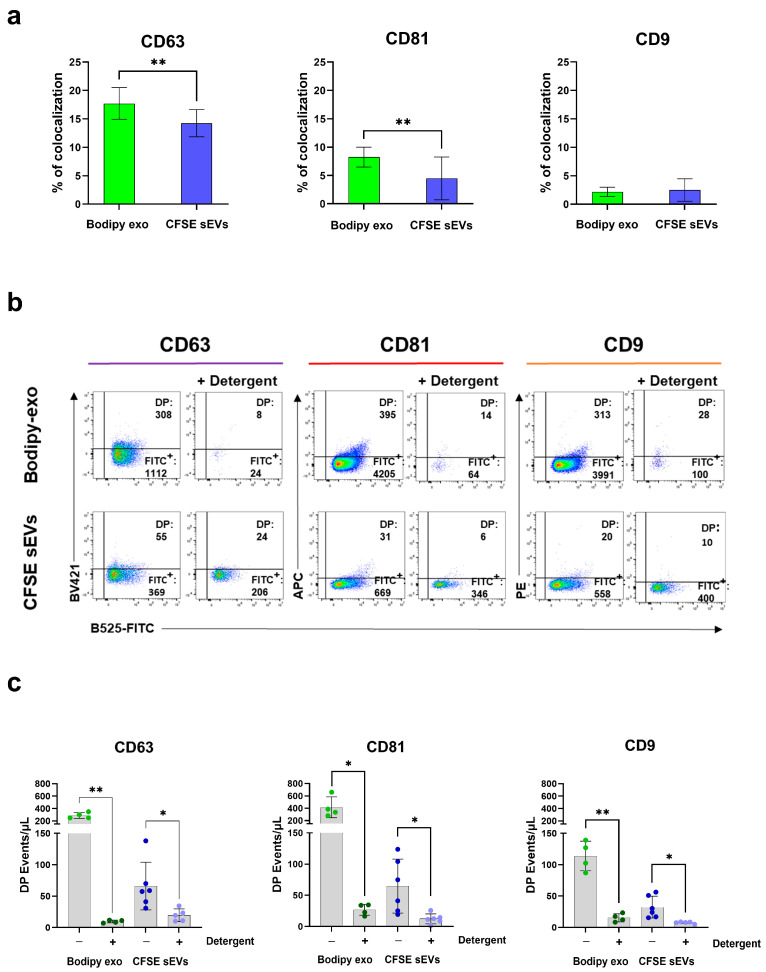
CFSE-stained vesicle analysis. (**a**) FC analysis of colocalization of tetraspanin fluorescent antibodies with Bodipy-exo compared to total CFSE-sEVs. To nonspecifically label the total sEVs population, 100 K pellets from conditioned media of unlabeled cells were incubated with CFSE. CD63 staining was performed in a 100 µL assay volume, while CD81 and CD9 staining were conducted in a 25 µL assay volume. (**b**) Dot plot analysis of Bodipy-exo and CFSE sEVs incubated with anti-CD63, anti-CD81, and anti-CD9 before and after treatment with the detergent Octyl-β-glucoside. Samples were acquired at a medium flow rate (30 μL/min), and the corresponding isotype controls were used to set the gating strategy. The events/μL in the double-positive (DP) and FITC-positive (FITC**^+^**) regions are reported. Data are representative of n ≥ 4 independent experiments. (**c**) Quantification of the DP events/μL before and after detergent lysis. Data are expressed as mean ± SD (n ≥ 4). Statistical analysis was performed using paired Student’s *t*-test. *p* values < 0.05 were considered statistically significant. * *p* < 0.05; ** *p* < 0.01.

**Figure 5 cells-14-00989-f005:**
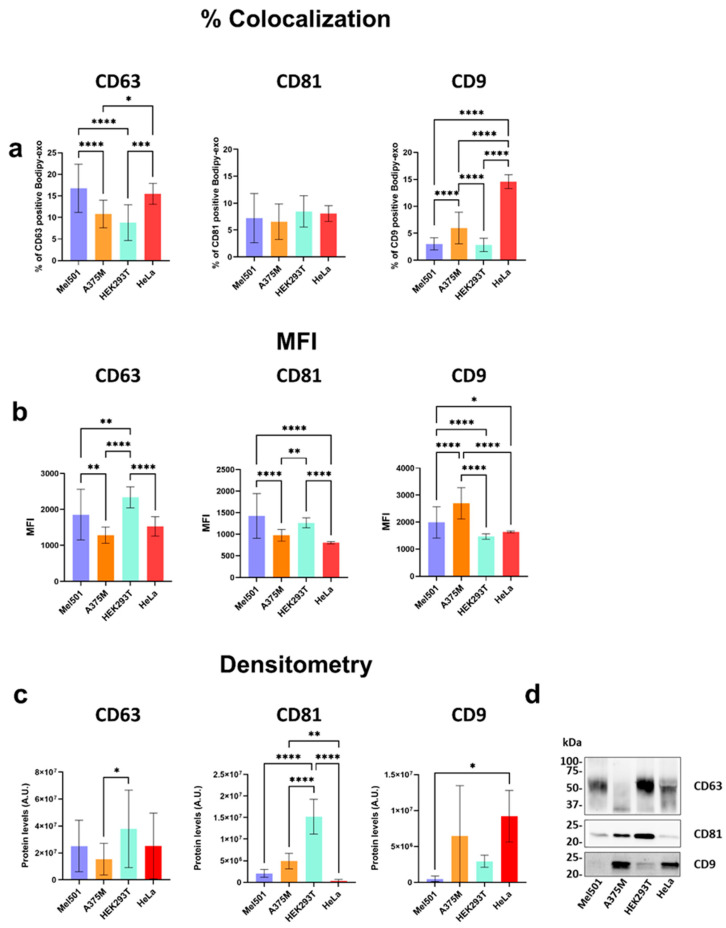
Heterogeneous distribution of tetraspanins on sEVs from different cell lines. (**a**) Percentage of tetraspanin-positive Bodipy-exo. (**b**) The mean fluorescence intensity (MFI) of each anti-tetraspanin antibody bound to Bodipy-exo was measured using the same instrument/acquisition gating settings applied for the quantification of Bodipy-exo samples. (**c**) Western blot analysis of sEVs released by different cell lines normalized for Bodipy-exo number. (**d**) Representative Western blot of at least three independent experiments. Data are expressed as mean ± SD (n ≥ 3). Statistical analysis was performed using one-way ANOVA and Tukey’s multiple comparison test. *p* values < 0.05 were considered statistically significant. * *p* < 0.05, ** *p* < 0.01, *** *p* < 0.001, and **** *p* < 0.0001.

## Data Availability

The data generated or analyzed during this study are either included in this article and its Appendix A or available from the corresponding author on reasonable request.

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
