# Peer review of "A Simple and Scalable Assay for Multiplexed Flow Cytometric Profiling of Surface Markers on Small Extracellular Vesicles"

_cells, 2025, doi:10.3390/cells14130989_

Round 1
Reviewer 1 Report
Comments and Suggestions for Authors
Please find comments in attachment.

Reviewer 2 Report
Comments and Suggestions for Authors
The authors aimed to use a simple and cost-effective flow cytometry-based assay for the multiplexed analysis of tetraspanin markers on fluorescently labeled sEVs. The study is well designed and well prepared. There are minor concerns that should be addressed.
- Abstract: What does CFSE mean?
- The introduction part should be divided into 2-3 paragraphs.
- The methods lack references
- How to verify the method is correct?
Author Response
We sincerely thank the reviewer for the encouraging comments on our manuscript. We greatly appreciate the time and effort invested in reviewing our work and the opportunity to address the minor concerns raised. Below, we provide point-by-point responses to the reviewer’s suggestions.
1.Abstract: What does CFSE mean?
We thank the reviewer for pointing this out. In the revised abstract, we have now defined CFSE at its first mention as carboxyfluorescein succinimidyl ester, a membrane-permeable fluorescent dye commonly used to label small extracellular vesicles (sEVs) (line 21).
2.The introduction part should be divided into 2-3 paragraphs.
We agree with the reviewer that breaking up the Introduction improves readability and flow. We have revised this section accordingly by dividing it into three paragraphs: (i) background on EVs (ii) challenges in current methods for EV protein profiling, and (iii) the rationale and aim of our study.
3.The methods lack references
We thank the reviewer for this important observation. In the revised manuscript, we have clarified how our protocol builds upon and extends our previously published work (Barreca et al., J Extracell Vesicles, 2023, Coscia et al. 2016). We carefully reviewed the Methods section and have now included appropriate references to support key procedures. We also added a missing citation at line 116. In addition, we would like to highlight that the combination of metabolic labeling with paraformaldehyde (PFA) fixation and low-speed centrifugation represents a novel aspect of our methodology. To our knowledge, no prior publication describes this specific workflow. PFA fixation, a standard technique in flow cytometry, was used to preserve vesicle integrity and surface antigenicity during immunostaining. We hypothesized that PFA-induced crosslinking increases vesicle rigidity, thereby improving sedimentation efficiency at lower centrifugal forces. This hypothesis was experimentally supported: by quantifying Bodipy-labeled sEVs before and after centrifugation, we demonstrated that PFA-fixed vesicles could be efficiently recovered using a benchtop centrifuge, with yields comparable to ultracentrifugation.
4.How to verify the method is correct?
This is an important point. Our method is verified through multiple validation steps, including the use of appropriate controls (isotype antibodies, PBS-only samples, and CFSE-stained vesicles) to ensure specificity and minimize background. Additionally, we used metabolically labeled sEVs that were previously validated as exosomes in our prior publication (Barreca et al., 2023). To further support our findings, we compared our flow cytometry data to Western blot analysis of CD63 across multiple cell lines (Figure 5d), demonstrating consistency between single-vesicle flow cytometry and bulk protein detection. These complementary approaches support the accuracy and robustness of the method.
